# Assessment of the Interactions Between Hemicellulose Xylan and Kaolinite Clay: Structural Characterization and Adsorptive Behavior

**DOI:** 10.3390/polym17141958

**Published:** 2025-07-17

**Authors:** Enzo Díaz, Leopoldo Gutiérrez, Elizabeth Elgueta, Dariela Núñez, Isabel Carrillo-Varela, Vicente A. Hernández

**Affiliations:** 1Facultad de Ciencias Químicas, Universidad de Concepción, Edmundo Larenas 129, Concepción 4070371, Chile; endiaz@udec.cl; 2Departamento de Ingeniería Metalúrgica, Universidad de Concepción, Edmundo Larenas 219, Concepción 4070371, Chile; lgutierrezb@udec.cl; 3Centro de Recursos Hídricos para la Agricultura y Minería CRHIAM, Av. Vicente Méndez 595, Chillán 3780000, Chile; 4Departamento de Química Ambiental, Facultad de Ciencias, Universidad Católica de la Santísima Concepción, Concepción 4070129, Chile; dnunez@ucsc.cl; 5Centro de Investigación en Biodiversidad y Ambientes Sustentables (CIBAS), Universidad Católica de la Santísima Concepción, Concepción 4070129, Chile; 6Centro de Investigación de Polímeros Avanzados (CIPA), Concepción 4051381, Chile; 7Bioforest SpA, Camino a Coronel Km 15, Coronel 4190000, Chile; isabel.carrillo@arauco.com; 8Departamento de Manejo de Bosques y Medio Ambiente, Facultad de Ciencias Forestales, Universidad de Concepción, Victoria 631, Concepción 40730386, Chile; 9Centro de Biotecnología, Universidad de Concepción, Barrio Universitario S/N, Concepción 40730386, Chile; 10Centro Nacional de Excelencia para la Industria de la Madera (CENAMAD), Pontificia Universidad Católica de Chile, Vicuña Mackenna 4860, Santiago 7820436, Chile

**Keywords:** xylan, kaolinite, clay, hemicellulose, flotation, mineral processing

## Abstract

In this study, a methacrylic derivative of xylan (XYLMA) was synthesized through transesterification reactions, with the aim of evaluating its physicochemical behavior and its interaction with kaolinite particles. Structural characterization by FT-IR and NMR spectroscopy confirmed the incorporation of methacrylic groups into the xylan (XYL) structure, with a degree of substitution of 0.67. Thermal analyses (TGA and DSC) showed a decrease in melting temperature and enthalpy in XYLMA compared to XYL, attributed to a loss of structural rigidity. Thermal analyses (TGA and DSC) revealed a decrease in the melting temperature and enthalpy of XYLMA compared to XYL, which is attributed to a loss of structural rigidity and a reduction in the crystalline order of the biopolymer. Aggregation tests in solution revealed that XYLMA exhibits amphiphilic behavior, forming micellar structures at a critical aggregation concentration (CAC) of 62 mg L^−1^. In adsorption studies on kaolinite, XYL showed greater affinity than XYLMA, especially at acidic pH, due to reduced electrostatic forces and a greater number of hydroxyl groups capable of forming hydrogen bonds with the mineral surface. In contrast, modification with methacrylic groups in XYLMA reduced its adsorption capacity, probably due to the formation of supramolecular aggregates. These results suggest that interactions between xylan and kaolinite clay are key to understanding the role that hemicelluloses play in increasing copper recovery when added to flotation cells during the processing of copper sulfide ores with high clay content.

## 1. Introduction

Copper is one of the most industrially important base metals worldwide, primarily used as an electrical conductor, especially in wiring. In the Earth’s crust, copper predominantly occurs in copper–iron sulfide ores such as chalcopyrite (CuFeS_2_) and chalcocite (Cu_2_S) [1,2,3,4]. It is estimated that approximately 80% of the world’s copper mineral resources are contained in Cu–Fe–S compounds, with chalcopyrite being the most abundant but also the most refractory, making it difficult to dissolve efficiently in aqueous solutions [5,6]. Due to this challenge, copper extraction from these ores is largely carried out through pyrometallurgical routes, which include flotation, smelting, and refining processes.

Prior to pyrometallurgical treatment, flotation is the primary concentration method responsible for generating significant quantities of tailings as by-products. According to the literature, 90% to 95% of the tailings produced in processing plants originate from flotation stages, including both primary and cleaning tailings [3,7,8]. Current copper production technologies generate approximately 19 million tons annually. However, despite the high production levels, there is growing concern over the environmental impacts associated with conventional processes, particularly sulfur dioxide (SO_2_) emissions released during smelting and refining [5,6]. This situation underscores the need to develop more sustainable and environmentally friendly hydrometallurgical alternatives.

One of the main limitations of conventional leaching of chalcopyrite under sulfide conditions is its low copper extraction rate, which undermines its large-scale commercial viability due to poor economic feasibility. This inefficiency is largely attributed to the formation of a passive layer on the mineral surface that inhibits copper dissolution [9,10]. Furthermore, copper sulfide ores with high clay content, such as chalcopyrite associated with kaolinite or montmorillonite, pose an additional challenge during flotation. The presence of clay minerals increases gangue hydrophobicity, promoting the entrainment of unwanted fine particles into the concentrate. Clays also consume flotation reagents (collectors and frothers), reducing their effectiveness for valuable minerals, and generate unstable or excessive froths that compromise concentrate recovery and grade. Additionally, clay-induced sliming produces ultrafine particles that hinder selective separation [11,12].

The use of natural polysaccharides as depressants in the flotation of copper ores with high clay content offers a sustainable and effective alternative for improving process selectivity and recovery. The choice of the appropriate polysaccharide will depend on the specific characteristics of the ore and the operating conditions of the plant [13,14,15,16]. For this reason, various natural polysaccharides have been investigated as depressants or dispersants to mitigate the adverse effects of clays. Fenugreek gum is particularly notable for its use as a depressant in the flotation of minerals with high clay content, where it interacts with minerals such as pyrophyllite through hydrogen bonds, reducing their hydrophobicity and improving process selectivity. Dextrin and starch derivatives are used as depressants for gangue minerals such as talc and clays, where they are adsorbed onto the surface of unwanted minerals, reducing their floatability. Guar gum and its derivatives are used as clay depressants in the flotation of copper minerals, interacting with clay and thus reducing the interference during flotation of valuable minerals [13,14,15].

Previous research has shown that the use of hemicelluloses as additives during flotation of high-clay copper sulfide ores can significantly improve copper recovery [17,18]. The action of hemicelluloses is related to their ability to interact with clay minerals. The semi-amorphous structure of hemicelluloses and the presence of hydroxyl groups in the side chain allows for their interaction with clay through hydrogen bonds, covering and neutralizing their surfaces. When clay particles adhere to hemicelluloses, their capacity to interfere with the flotation of copper sulfides is significantly diminished. Accordingly, the objective of this study was to investigate the interactions between xylan hemicellulose, extracted from eucalyptus bleached kraft pulp, and the kaolinite mineral, which is commonly found in high-clay copper sulfide ores [15,19,20]. Building on this principle, the aim of derivatizing xylan with methacrylic acid is to introduce unsaturated methacrylic groups into the hemicellulose backbone, thereby transforming it into a reactive, amphiphilic biopolymer with tailored functional sites. This structural modification enables further crosslinking or polymerization, expands its capacity to form hydrogels, micelles, or composite networks, and modulates key properties such as crystallinity, solubility, and interfacial interactions.

Hemicelluloses are abundant in many agricultural waste and forest products, including wheat, barley, oats, wood waste products. They are made up of sugar units linked by glycosidic bonds formed by more than one type of sugar, such as hexoses or pentoses, with different branches and substitutions. In the cell wall, they serve as a bridge between cellulose and lignin, and act as a support material [15,18,21]. In their natural state, they can exist in amorphous form with a low degree of polymerization of only 200 monomers. However, despite their high potential and abundancy, hemicelluloses are mainly ignored in the forestry and agricultural industries, being either burned or disposed as organic waste [22,23]. Hemicelluloses can be divided into four general classes of polysaccharides: xylans, mannans, β-glucans with mixed linkages, and xyloglucans. In hardwood species, the main hemicellulose type is xylan, which predominantly consists of D-xylopyranoside units connected by β-(1→4) linkages with glucuronic acid and acetyl groups [24]. Xylan can be extracted from biomass by alkali or autohydrolysis, and it can be conveniently modified by targeting its hydroxyl groups. Among the most important reported modifications of xylan are etherification with epoxides and alkyl halides, the development of bifunctional derivatives using acrylamide and acrylic acid, and esterification employing anhydrides or activated carboxylic acids [24,25,26].

The objective of this work was to study the interactions between xylan hemicellulose extracted from eucalyptus bleached kraft pulp, a forest product, and kaolinite mineral, widely present in high clay copper sulfide ores. These interactions were evaluated under different pH conditions and a varying xylan molecular structure. The molecular structure of xylan was modified with methacrylate to reduce the amount of hydroxyl groups available to interact with kaolinite. The structural characterizations of methacrylate-modified (XYLMA) and unmodified xylan (XYL) were conducted via nuclear magnetic resonance (NMR) and Fourier transform infrared spectroscopy (FTIR). In addition to this, the critical aggregation concentration (CAC) was determined by fluorescence spectroscopy, minimizing the influence of micelle formation during the absorption of the hemicellulosic polymer in kaolinite.

Therefore, understanding the interactions between these polysaccharides and clay under different pH conditions is of great interest for optimizing copper mineral recovery processes.

## 2. Materials and Methods

### 2.1. Materials

The analytical-grade reagents dimethyl sulfoxide (DMSO), dimethylaminopyridine (DMAP), glycidyl methacrylate (GMC), ethanol, deuterated dimethyl sulfoxide (DMSO-d6), tetramethylsilane (TMS), sodium hydroxide (NaOH), hydrochloric acid (HCl), and potassium hydroxide (KOH) were all obtained from Merck (Darmstadt, Hesse, Germany) and used without further purification. The XYL samples were extracted from bleached eucalyptus kraft pulp, and their number- and weight-average molecular weight were 3560 and 4125 Dalton, respectively. Kaolinite was obtained from the Clay Minerals Society (Chantilly, VA, USA). According to the supplier, the composition of the mineral, determined by XRD analyses, was 97% of kaolinite, with minor amounts of anatase and other impurities. The cation exchange capacity (CEC) and Brunauer–Emmett–Teller (BET) surface area were 2.0 meq/100 g and 10.05 m^2^ g^−1^, respectively. The kaolinite average particle size was 8.9 μm, and 100% of the particles had a size under 21 μm, as described by Ramírez et al. (2024) [27].

### 2.2. Extraction of Xylan Hemicelluloses

Xylan hemicellulose was extracted and precipitated from bleached eucalyptus kraft pulp provided by a local pulp mill in the Biobío Region of Southern Chile. Xylan was obtained through a selective extraction method involving a cold alkaline treatment followed by methanol precipitation. Initially, the bleached eucalyptus kraft pulp sheets were hydrated for 4 h, stirred for 30 min, centrifuged, and pelleted. For the alkaline extraction, the pelleted pulp (with 59% moisture content) was treated with 5% KOH and stirred for 2 h in an Erlenmeyer flask to solubilize the hemicellulose. The mixture was then vacuum-filtered, and the resulting liquid was neutralized with HCl to a pH of 7. Methanol was subsequently added, and the solution was left to stand for 16 h to promote xylan precipitation. Finally, the precipitate was concentrated, centrifuged, and freeze-dried.

### 2.3. Xylan Modification

Xylan was chemically modified following the method reported by De Smedt et al. and Cao et al. to produce XYLMA [28,29] (Figure 1). Modification with methacrylate (MA) was achieved by mixing 0.50 g of xylan with 30 mL of DMSO in a 100 mL round flask. The mixture was heated to 95 °C under stirring for 1.5 h. After cooling to room temperature, 0.1 g of 4-dimethylaminopyridine (DMAP) was added as a catalyst, and the solution was kept under stirring at 40 °C for another 30 min, after which 0.54 g of glycidyl methacrylate (GMA) was added. Then, stirring continued for 36 h at 40 °C. After that, the reaction mixture was allowed to cool to room temperature, and 80 mL of 95% ethanol was added under constant stirring. Finally, the pale-yellow precipitate obtained in the process was filtered, washed with an abundance of ethanol, and dried in a vacuum oven to obtain the XYLMA sample.

### 2.4. XYL and XYLMA Characterizations

Both ^1^H and ^13^C (DEPT-135) NMR spectra of XYL and XYLMA samples were obtained using a Bruker Avance 400 MHz, Ascend TM model spectrometer (Billerica, MA, USA), with DMSO–d6 as a solvent and trimethylsilane (TMS) as an internal standard. The FTIR spectra of both polymers was recorded with a Nicolet Magna 550 spectrometer (Nicolet Instrument Corp., Madison, WI, USA) in a spectral range of 4000 to 400 cm^−1^ on a KBr disk. Thermogravimetric analysis (TGA) was performed on a NETZSCH 209 F1 Iris model (TA Instruments, Burlington, VT, USA), and differential scanning calorimetry (DSC) analysis was performed on a NETZSCH 204 F1 Phoenix (Burlington, VT, USA) at a heating rate of 10 °C min^−1^.

The CAC of xylan was measured at pH 3 through steady-state fluorescence spectroscopy using pyrene as a fluorescent probe. A saturated aqueous solution of pyrene was used to prepare solutions of XYL or XYLMA at different concentrations, ranging from 0.007 to 0.700 mg mL^−1^. Excitation spectra of pyrene were acquired by using a Quanta Master fluorescence spectrometer (Photon Technology International Inc., Birmingham, AL, USA) in a range between 300 and 370 nm, with an emission wavelength of 383 nm. The CACs of both xylan polymers were obtained from a plot of the fluorescence intensity ratio of 337/334 nm versus the logarithm of polymer concentration (SCHIMADZU, Kyoto, Japan) [30].

### 2.5. Adsorption Tests

The adsorption of hemicelluloses on kaolinite was measured through tests in which 0.5 g of kaolinite was mixed with 50 mL of XYL or XYLMA solutions prepared in nano-pure water at known concentrations in a 100 mL flask. The suspensions were kept under stirring at 500 rpm for 24 h at 25 °C, and then centrifuged for 5 min at 7000 rpm. The obtained supernatants were analyzed for total organic carbon (TOC) in a Shimadzu TOC-VCPH analyzer (Shimadzu Corporation, Kyoto, Japan). The TOC readings were translated into XYL or XYLMA concentrations through a calibration curve. The mass of the adsorbed hemicellulose on kaolinite was calculated as the difference between the concentrations in the initial and final solutions multiplied by the volume of the solution (50 mL). Then, the specific adsorption was calculated as milligrams of adsorbed XYL or XYLMA per gram of kaolinite (mg g^−1^). Temperature was controlled during the experiments using a thermostatic bath. The tests were conducted in duplicates, with an average standard error of 4%. The experiments were conducted at different pH values (2, 4, 6, 8, and 10) adjusted using KOH and HCl solutions.

## 3. Results and Discussion

### 3.1. XYL and XYLMA Characterizations

Figure 2 shows the FTIR (a) and ^1^H NMR (b) spectra of the XYL and XYLMA samples, respectively. The FTIR spectrum of XYLMA exhibits a pronounced peak at 1719 cm^−1^, confirming the incorporation of carbonyl groups, attributed to methacrylic functionalization. Additionally, the absorption bands detected at 2927 cm^−1^ and 3441 cm^−1^ are assigned to the stretching vibrations of C–H bonds and hydroxyl groups, respectively. The NMR spectra of ^1^H also showed signals at 1.9, 5.6, and 6.0 ppm indicating the insertion of methacrylic groups in XYLMA. The signal at 1.9 ppm corresponds to the protons of the methyl group and the signals at 5.6 ppm and 6.0 ppm correspond to the protons of the vinyl group. Figure 3 shows the ^13^C NMR (a) and DEPT-135 NMR (b) spectrum of the XYLMA polymer. The ^13^C NMR spectrum (see Figure 3a) shows the CH_3_ group at 19 ppm, CH_2_ from the vinyl group (C=C) at 125 and 136 ppm, and the presence of carboxylic carbon (C=O) at 166 ppm. The signals between 60 and 110 ppm correspond to the carbons of the xylose units. The septet signal at 39.52 ppm, which corresponds to the DMSO-d6 solvent, was removed from the graph to improve resolution. Figure 3b shows the DEPT-135 NMR spectrum. This spectrum indicates the presence of the CH_3_ group (positive signal) of the methacrylic group at 18 ppm, the presence of CH_2_ from vinyl at 126 ppm (a negative signal), and the signal corresponding to the CH_2_ group of xylose units at 63 ppm. Therefore, the spectral analysis of XYLMA showed that the reaction of XYL with GMC resulted in the corresponding derivatives via a transesterification reaction, where methacrylic groups were inserted into the structure of the XYL molecule. Additionally, similar results were obtained by Peng et al. [31] and Elgueta et al. [32] in the insertion of this same molecular fragment in xylan extracted from wood. Through the integration of the signals in the ^1^H NMR spectrum and by applying Equation (1), based on the methodology reported by Lowman [33,34] and Kallakas et al. [35], it was possible to determine that the degree of substitution (DS) in XILMA is 0.67. This value is indicative that, on average, there are two units of MA inserted for every three units of xylose present, approximately, in the polysaccharide.DS = 8 × ((ICH_3_))/(3 × (IAUX) + (ICH_3_))(1)

In Equation (1), (ICH_3_) corresponds to the integral of the signal of the CH_3_ group of the MA unit, (IAUX) corresponds to the integral of the signal of the anhydrous xylose unit (AUX), 8 corresponds to the number of protons present in the anhydrous xylose unit, and 3 corresponds to the number of protons present in the methyl group of the MA unit.

The thermal properties of XYL and XYLMA were studied via TGA and DSC. The thermograms are shown in Figure 4a,b. The XYL and XYLMA compounds undergo decomposition by 50% at about 280 °C and decomposition between 80 and 90% (final temperature of the analysis, see Figure 4a) at 550 °C. Figure 4b shows the DSC analysis, where it can be observed that XYL has a melting temperature of 110 °C with an enthalpy of 292 J g^−1^ and XYLMA has a melting temperature of 72 °C with an enthalpy of 58 J g^−1^. It should be noted that the FTIR spectrum of XYLMA shows an increase in the intensity of the band corresponding to hydroxyl groups in relation to XYL, which at first glance might seem contradictory to the decrease observed in melting temperature and enthalpy. This apparent discrepancy can be attributed to the partial disruption of the dense network of hydrogen bonds characteristic of native hemicellulose. Methacrylic functionalization breaks certain intermolecular bonds and introduces bulky carbonyl groups that generate steric hindrances, hindering the orderly packing of polymer chains and the formation of crystalline domains. As a result, a greater number of exposed hydroxyl groups are detected spectroscopically, although these participate less effectively in structuring hydrogen bonds. Likewise, the increase in structural disorder and the amorphous fraction favors greater chain mobility, which translates into lower thermal stability. Consequently, despite the increase in detectable hydroxyl groups, the overall efficiency of hydrogen bonds and the degree of crystalline order are reduced, as evidenced by lower melting temperature and enthalpy values [31,36,37].

The CAC measurements showed that the XYL sample did not display aggregation at the concentration range tested, presenting highly polar behavior and solubility in water. In contrast, the CAC of XYLMA (Figure 5) revealed that the intensity of the excitation spectrum of pyrene decreased at high concentrations of XYLMA, with a shift from approximately 328 nm to 330 nm. The CAC was determined from the plot of I328/I330 versus the logarithm of polymer concentration, as shown in Figure 6. The results indicated that XYLMA was able to form micelles at a concentration as low as 0.062 mg mL^−1^. This result is in agreement with findings from the related research in the area, which shows that polysaccharides are often limited to a secondary structure level, while hydrophobically modified polysaccharides (HMPs) can give rise to innumerable possibilities to generate complex architectures at a three-dimensional structural level, including micelles, polymersomes, reverse micelles and hydrogels [38]. The incorporation of hydrophobic substituents in polysaccharides provides a greater diversity of intermolecular forces, which is beneficial for the stability of supramolecular structures formed in aqueous solutions.

### 3.2. Adsorption of XYL and XYLMA on Kaolinite

Figure 7 shows the results of specific adsorptions of XYL and XYLMA on kaolinite particles at various pH values. The results showed that the specific adsorption of XYL and XYLMA on kaolinite increased as the pH was decreased, and that in highly alkaline conditions, the adsorption of both polymers tended to be similar. To understand the high adsorption of both xylan polymers at acidic pH values, it was necessary to consider the surface properties of kaolinite and the molecular behavior of xylan. Kaolinite is a clay mineral that has been proposed to take to two different surfaces created during fracture or particle breakage. This means that the basal silica-like faces that are negatively charged over a wide pH range, and the alumina-like edges that are positively charged in the neutral-acidic pH range indicate a high face/edge ratio [39,40]. Previous results showed that silica-like faces are negatively charged at pH > 4, and that alumina-like edges are positively charged at pH < 6, and negatively charged at pH > 8 [41]. It was also proposed that basal plane surfaces or faces of kaolinite should display some hydrophobicity, similarly to other phyllosilicates such as pyrophyllite and talc [42]. However, the fact that kaolinite particles are easily dispersed in water has been explained by the presence of surface defects on the basal planes which create polar micro-edges and a patch-wise surface heterogeneity [43,44]. Conversely, XYL molecules have carboxylic groups in their structures, which are deprotonated at high pH values, thus causing negative charges in their structure [45]. At low pH values, these carboxylic groups are protonated leaving the XYL structures with no charge. Consequently, in our study, at lower pH values (2 to 3), both the kaolinite surfaces (mostly faces) and XYL charges were reduced, and this reduction in electrostatic repulsion facilitated the adsorption of XYL on kaolinite, most probably due to hydrogen bond interactions between hydroxyl groups present in the xylan and silanol groups of kaolinite. Figure 8 shows a simplified representation of the interaction enabled by hydrogen bridges between XYL and kaolinite. The results presented in Figure 8 also show that, in the pH range of 2 to 4, the adsorption of XYL on kaolinite particles was approximately two-fold higher than the adsorption of XYLMA, and that this difference was substantially reduced with increasing pH values. These results can be explained by the modification of the XYL structure with methacrylic groups, which reduced the amount of hydroxyl groups available in the xylan molecule. As a result, the hydrogen bond interactions and the specific adsorption of XYLMA on kaolinite were expected to be reduced even though electrostatic interactions decreased at low pH values.

Figure 9 shows the results of specific adsorptions of XYL and XYLMA on kaolinite at various concentrations and at pH 3. These tests were performed at pH 3 to reduce the interference of electrostatic interactions between polymers and minerals. The results showed that the adsorption of XYL on kaolinite increased with concentration, reaching a maximum of 69.2 mg g^−1^ at a concentration of 800 mg L^−1^. Since no plateau was observed, it can be proposed that, in the range of concentrations tested, the adsorption of XYL on kaolinite took place through cooperative effects in which the molecules that were already adsorbed enhanced the adsorption of more molecules. This significantly lower adsorption of XYLMA on kaolinite compared to that of XYL can be explained by two main factors: the reduction in hydrogen bonding capacity and the tendency of XYLMA to self-aggregate. Methacrylic functionalization introduces bulky carbonyl groups that partially block hydroxyl sites, thereby decreasing the number of sites available for hydrogen bonding with the kaolinite surface. At the same time, the amphiphilic nature of XYLMA promotes intra- and intermolecular interactions, leading to the formation of supramolecular aggregates in solution. These aggregates reduce the ability of individual polymer chains to interact with the mineral surface, further limiting adsorption. Therefore, both the decrease in hydrogen bonding capacity and the self-aggregation behavior of XYLMA contribute synergistically to its lower adsorption compared to that of unmodified XYL [31,46,47].

In contrast, it was found that the adsorption of XYLMA on kaolinite reached a plateau at around the CAC, and it was noticeably lower than the adsorption of XYL at all the concentrations tested. As previously explained, such differences may be due to the smaller number of hydroxyl groups available in XYLMA and the presence of methacrylic groups that increase the hydrophobicity of the polymer, thus enhancing its amphiphilic properties. Formation of XYLMA supramolecular aggregates diminishes their adsorption on the surface of kaolinite. This can be observed at concentrations between 50 mg L^−1^ and 100 mg L^−1^, where an inflection occurs in the isotherm, and over this inflection, the adsorbed amount remains relatively constant. In this range, the CAC was located, and its value was determined as 62 mg L^−1^. The formation of micelles reduced the exposed surface area of XYLMA molecules, hence decreasing their possible interaction with kaolinite.

Further research on the interactions between hemicelluloses and clay minerals may include the synthesis of new hemicellulose derivatives to be tested on kaolinite and other phyllosilicates. These compounds might incorporate ionic groups, such as sulfonates and ammonium salts in their structure, to test other types of interaction between the organic compounds and the minerals.

## 4. Conclusions

In conclusion, this study confirmed the successful modification of xylan (XYL) through the insertion of methacrylic groups, resulting in the derivative XYLMA, as demonstrated by FT-IR and NMR (^1^H, ^13^C, and DEPT-135) spectroscopic analyses. Notably, the FTIR spectrum of XYLMA showed an increase in the intensity of the hydroxyl band compared to XYL, which at first glance might appear inconsistent with the observed decrease in melting temperature and enthalpy. This apparent contradiction can be explained by the partial disruption of the dense hydrogen-bonded network characteristic of native hemicellulose, where the introduction of bulky carbonyl groups reduces the effectiveness of intermolecular hydrogen bonding and promotes greater chain mobility. The incorporation of these functional groups significantly altered the thermal and structural properties of the biopolymer, as evidenced by the lower melting temperature and enthalpy observed in TGA and DSC analyses. Additionally, this functionalization imparted amphiphilic properties to XYLMA, enabling the formation of micellar structures in aqueous solution at a critical aggregation concentration (CAC) of 62 mg L^−1^, a behavior not observed in unmodified xylan.

Regarding mineral interactions, XYL exhibited a higher adsorption capacity on kaolinite, especially under acidic conditions, attributed to the greater availability of hydroxyl groups capable of forming hydrogen bonds with the silanol groups on the clay surface. In contrast, XYLMA showed lower adsorption, likely due to steric hindrance, reduced hydrogen bonding efficiency, and the formation of supramolecular aggregates that limited its active surface area. Taken together, these results demonstrate how the hydrophobic functionalization of hemicelluloses, derived as by-products from the wood industry, can modulate their self-assembly behavior and interactions with minerals, opening up new opportunities for applications in colloidal systems, encapsulation, adsorption processes, and the development of composite materials.

## Figures and Tables

**Figure 1 polymers-17-01958-f001:**
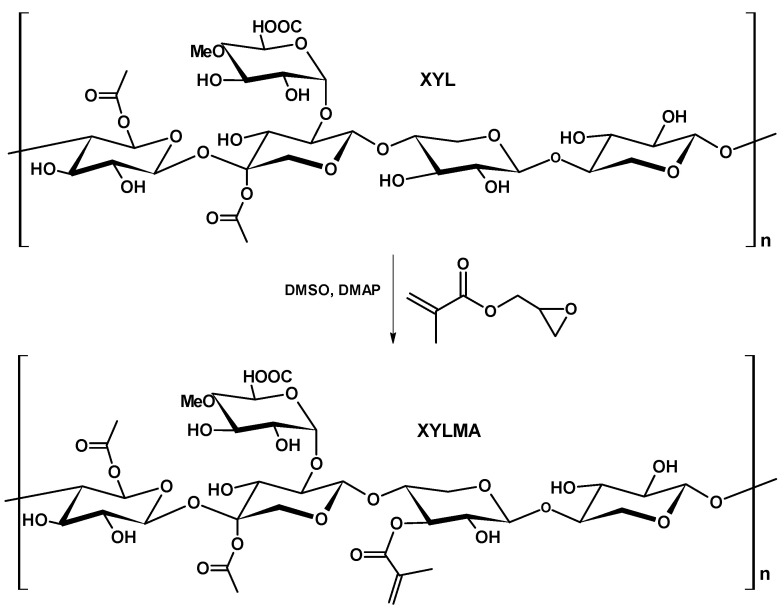
XYLMA synthesis scheme.

**Figure 2 polymers-17-01958-f002:**
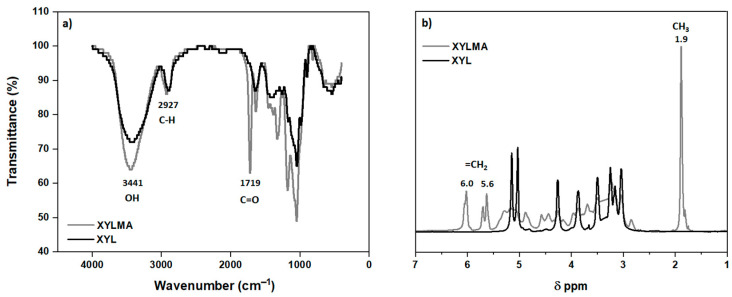
(**a**) FT-IR spectra for XYL and XYLMA; (**b**) ^1^H NMR spectra for XYL and XYLMA.

**Figure 3 polymers-17-01958-f003:**
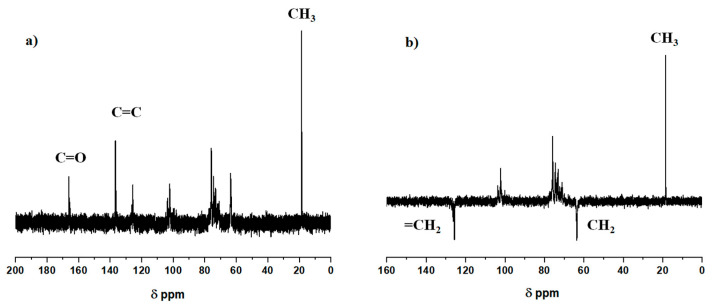
(**a**) ^13^C NMR and (**b**) DEPT 135 NMR spectrum for XYLMA.

**Figure 4 polymers-17-01958-f004:**
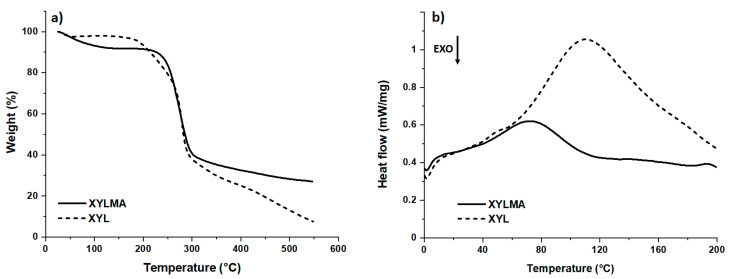
Thermogravimetric analysis (**a**) and differential scanning calorimetry (**b**) of XYL and XYLMA.

**Figure 5 polymers-17-01958-f005:**
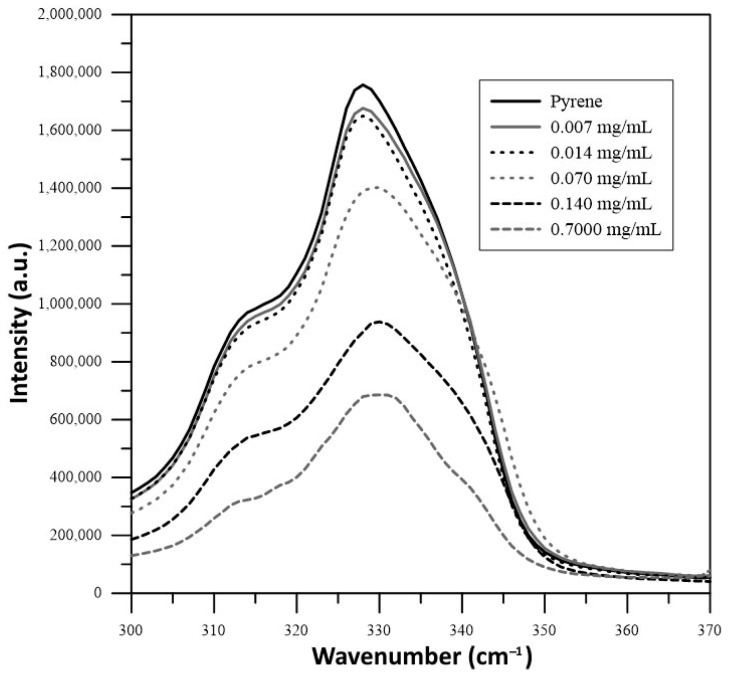
Excitation spectrum of pyrene in solutions containing increasing concentrations of XYLMA.

**Figure 6 polymers-17-01958-f006:**
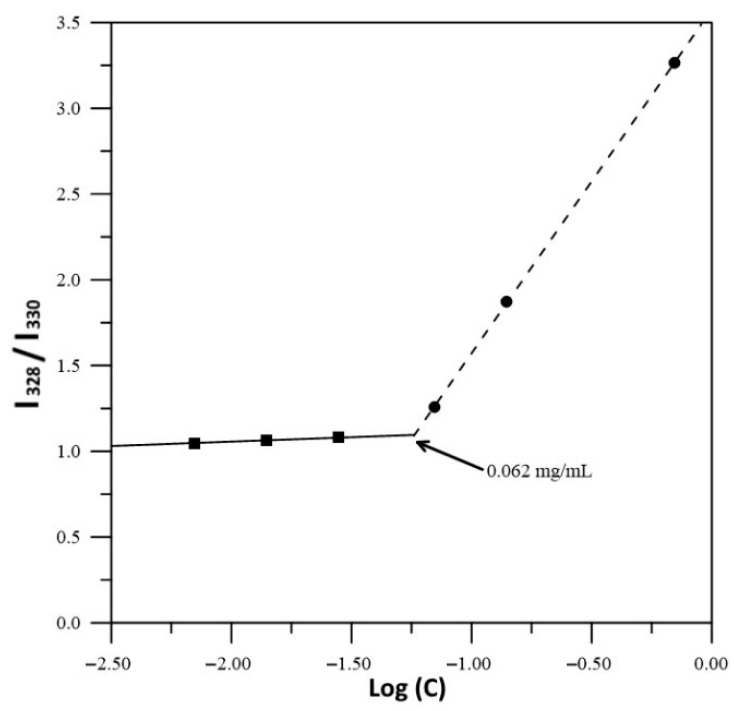
Dependence of the intensity ratio (I328/I330) of pyrene with the concentration of XYLMA (Log (C)).

**Figure 7 polymers-17-01958-f007:**
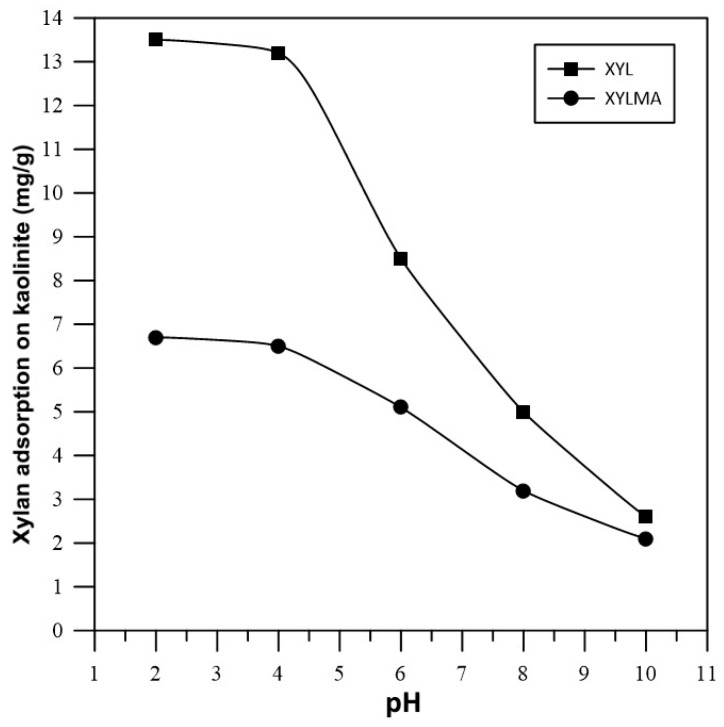
Adsorptions of XYL and XYLMA on kaolinite at various pH values; initial concentration, 0.7 mg mL^−1^.

**Figure 8 polymers-17-01958-f008:**
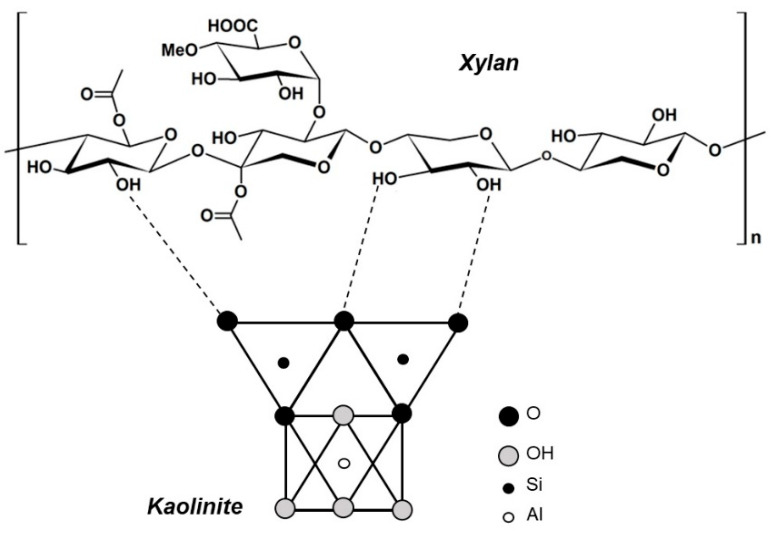
Simplified representation of the hydrogen bond interactions between xylan hemicellulose and kaolinite.

**Figure 9 polymers-17-01958-f009:**
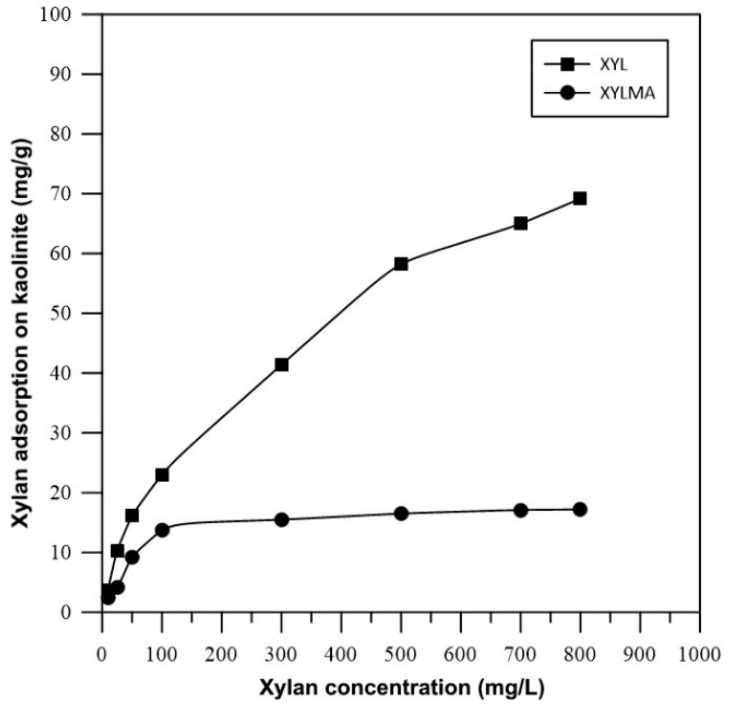
Adsorptions of XYL and XYLMA on kaolinite at various concentrations of xylan and at pH 3.

## Data Availability

All data from this study are available upon request.

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
