# Peer review of "Assessment of the Interactions Between Hemicellulose Xylan and Kaolinite Clay: Structural Characterization and Adsorptive Behavior"

_polymers, 2025, doi:10.3390/polym17141958_

Round 1
Reviewer 1 Report
Comments and Suggestions for Authors
See the attached file.

Author Response
We thank very much the comments, corrections, and suggestions of the referees. We send a new version of the paper which includes the modifications made on the manuscript according the correction referee’s. The following responses to the referee’s comments are given and are marked in red color at the revised version.
Please, find enclosed the revised version of the manuscript polymers-3738003 entitled “Assessment of the interactions between hemicellulose Xylan and kaolinite clay: Structural characterization and adsorptive behavior” by Enzo Díaz, Leopoldo Gutiérrez, Elizabeth Elgueta, Dariela Núñez, Isabel Carrillo-Varela and Vicente A. Hernández submitted to Polymers.
Comments and Suggestions for Authors
Editor 1
Diaz and coauthors studied interactions between hemicellulose Xylan and kaolinite clay. The methacrylic derivative of xylan was synthesized and characterized by FT-IR, NMR, TGA, and DSC. The adsorption studies on kaolinite indicated that XYL has a greater affinity than XYLMA. The author contributed the reduction of the methacrylic groups modified XYL to the formation of supramolecular aggregates in XYLMA. The reviewer agrees that the results are sufficient to support the conclusion. The reviewer would recommend publishing in the journal after addressing the comments below.
1. The experimental details (instrumentation, sample preparation, experimental parameters) of DSC and TGA need to be added in the Materials and Methods section.
R: Thank you for your observation. It was incorporated in the new version.
2. Figs. 2 and 3: The author should label the peaks in the graph that were mentioned in the text.
R: Thank you for your observation. It was incorporated in the new version.
3. Fig 5: The DSC curve should include exo/endo up or down information.
Thank you for your observation. It was corrected in the new version.
4. Fig 10: Can the author comment on the contribution of XYLMA aggregation to the reduction of its adsorption compared with the contribution from the reduced H-bonding?
Thank you for your comments, they were incorporated into the new version of the manuscript.
Reviewer 2 Report
Comments and Suggestions for Authors
This manuscript entitled “Assessment of the interactions between hemicellulose Xylan and kaolinite clay: Structural characterization and adsorptive behavior” aimed to study the interactions between xylan and kaolinite mineral. In my opinion, this work is meaningful and worth carrying out detailedly.
The present paper deals with a topic of potential interest for the readers of this journal. Nonetheless, there are points that should be addressed before re-considering it for publication. A possible list follows.
- The English in several sections requires a serious polishing.For example, Abstract: “Thermal analyses (TGA and DSC) showed a decrease in melting temperature and enthalpy in XYLMA compared to XYL,” It should be ----compared to that of XYL. Introduction: “By adhering to the hemicelluloses, they decrease their ability to interfere with the flotation of copper sulfides,”, “The objective of this work was to study the interactions between xylan hemicellulose extracted from eucalyptus bleached kraft pulp, a forest product, and kaolinite mineral widely present in high clay copper sulfide ores.” There are grammatical mistake in these sentence. Please amend the similar faults in all the text.
- Introduction: Page 2: “the presence of hydroxyl groups in the side chain, has been considered relevant to explain their interactions with clays through hydrogen bonds, covering or neutralizing their surfaces, which reduces their interference.”And XYL showed greater affinity than XYLMA, especially at acidic pH. Thus, the hydroxyl groups of Xylan is important in adhering with kaolinite clay. Please explain the aim or meaning of xylan derivatization by methacrylic acid.
- The logic of the introduction needs to be strengthened, especially for the first and second paragraph.
- Results and discussion: Figure 2 listed the transmittance. However, the peak in about 3400cm-1 of XYLMA was enhanced,please explain. Section 3.1 Fig 2,3, and 4 can be combined.
- Results and discussion: Page 7 “This effect probably occurs since hemicellulose has a three-dimensional structure composed of chemical bonds which give it greater thermal resistance and rigidity.”The reason need to be verified.
Author Response
We thank very much the comments, corrections, and suggestions of the referees. We send a new version of the paper which includes the modifications made on the manuscript according the correction referee’s. The following responses to the referee’s comments are given and are marked in red color at the revised version.
Please, find enclosed the revised version of the manuscript polymers-3738003 entitled “Assessment of the interactions between hemicellulose Xylan and kaolinite clay: Structural characterization and adsorptive behavior” by Enzo Díaz, Leopoldo Gutiérrez, Elizabeth Elgueta, Dariela Núñez, Isabel Carrillo-Varela and Vicente A. Hernández submitted to Polymers.
Comments and Suggestions for Authors
Editor 2
This manuscript entitled “Assessment of the interactions between hemicellulose Xylan and kaolinite clay: Structural characterization and adsorptive behavior” aimed to study the interactions between xylan and kaolinite mineral. In my opinion, this work is meaningful and worth carrying out detailedly.
The present paper deals with a topic of potential interest for the readers of this journal. Nonetheless, there are points that should be addressed before re-considering it for publication. A possible list follows.
1. The English in several sections requires a serious polishing. For example, Abstract: “Thermal analyses (TGA and DSC) showed a decrease in melting temperature and enthalpy in XYLMA compared to XYL,” It should be ----compared to that of XYL. Introduction: “By adhering to the hemicelluloses, they decrease their ability to interfere with the flotation of copper sulfides,”, “The objective of this work was to study the interactions between xylan hemicellulose extracted from eucalyptus bleached kraft pulp, a forest product, and kaolinite mineral widely present in high clay copper sulfide ores.” There are grammatical mistake in these sentence. Please amend the similar faults in all the text.
Thank you for your comments, the English writing has been revised and polished in the new version of the manuscript.
2. Introduction: Page 2: “the presence of hydroxyl groups in the side chain, has been considered relevant to explain their interactions with clays through hydrogen bonds, covering or neutralizing their surfaces, which reduces their interference.”And XYL showed greater affinity than XYLMA, especially at acidic pH. Thus, the hydroxyl groups of Xylan is important in adhering with kaolinite clay. Please explain the aim or meaning of xylan derivatization by methacrylic acid.
Thank you for your suggestions, these were integrated into the new version of the manuscript.
3. The logic of the introduction needs to be strengthened, especially for the first and second paragraph.
Thank you for your observation. It was corrected in the new version.
4. Results and discussion: Figure 2 listed the transmittance. However, the peak in about 3400cm-1 of XYLMA was enhanced, please explain. Section 3.1 Fig 2,3, and 4 can be combined.
Thank you for your comments, they were incorporated into the new version of the manuscript.
5. Results and discussion: Page 7 “This effect probably occurs since hemicellulose has a three-dimensional structure composed of chemical bonds which give it greater thermal resistance and rigidity.” The reason need to be verified.
Thank you for your suggestions, these were discussed in the new version of the manuscript.